

# Comparison of diversity and composition of macrofungal species between intensive mushroom harvesting and non-harvesting areas in Oaxaca, Mexico

Carolina Ruiz-Almenara[1,*], Etelvina Gándara[2,*] and
Marko Gómez-Hernández[3]

[1] CIIDIR Unidad Oaxaca, Instituto Politécnico Nacional, Santa Cruz Xoxocotlán, Oaxaca, Mexico
[2] Facultad de Ciencias Biológicas, Benemerita Universidad Autónoma de Puebla, Puebla, Puebla, Mexico
[3] CONACYT-CIIDIR Unidad Oaxaca, Instituto Politécnico Nacional, Santa Cruz Xoxocotlán, Oaxaca, Mexico

* These authors contributed equally to this work.

Corresponding author
Marko Gómez-Hernández,
magomezhe@conacyt.mx

## ABSTRACT

Wild edible mushrooms have been collected and consumed by human groups for centuries, and today they represent a relevant source of food and income for many rural families worldwide. Preserving these non-timber forest products is of great interest, and there is concern about the damage caused by intensive mushroom harvesting on macromycete communities. The aim of this study was to evaluate variation in diversity and composition of macromycete species between areas regularly used for mushroom harvesting and non-harvested areas in the Mixteca region of Oaxaca, Mexico, as well as to assess the influence of microclimatic and environmental factors on this variation. We selected two harvested and two non-harvested sites within the study area. In each one, we established 10 permanent plots of 10 m × 10 m where we sampled all the observed fruit bodies weekly from June to October 2017. We recorded a total of 856 individuals corresponding to 138 species, and 23 of these were identified as edible. Overall macromycete diversity, edible species diversity and composition were similar in Sites 1 (non-harvested) and 3 (harvested), and in Sites 2 (non-harvested) and 4 (harvested). Variation of diversity and species composition along the studied area was mainly related to microclimatic variables, while most environmental variables and variables related to vegetation structure similarly affected macromycete species in the four sites. Our results indicate that intensive harvesting of wild edible mushrooms is not affecting the diversity and distribution of macromycete species in our study area. Knowledge on the sustainability of mushroom harvesting practices can help improve current regulations regarding the management of these valuable non-timber forest products.

## INTRODUCTION

Fungi are of great importance in forest ecosystems worldwide. As decomposers, they are the most important organisms for the degradation of organic matter, and play a key role in nutrient cycling (*Lodge, 1993*; *Deacon, 2006*). Mycorrhizal fungi form symbiotic associations with higher plants, facilitating plant uptake of water and nutrients such as phosphorus and nitrogen, in exchange for photosynthetically fixed carbon (*Hall, Yun & Amicucci, 2003*; *Egli, 2011*). Plant and animal pathogenic fungi impact ecosystems mainly by acting as natural population regulators, thereby influencing productivity and species diversity and composition (*Hansen & Stone, 2005*; *Deacon, 2006*).

In addition to their roles in ecosystem functioning, fungi are highly relevant for humans and human-related activities (*Mueller, Bills & Foster, 2004*). Wild edible mushrooms have been collected and consumed by people for thousands of years and, given their nutritional value, some species are used as substitutes of meat in developing countries (*Boa, 2004*). Wild edible macromycetes are also among the most important non-timber forest products sold worldwide, generating ca. US$2 billion each year (*Boa, 2004*; *Voces, Diaz-Balteiro & Alfranca, 2012*). Information compiled from 10 countries revealed 2,166 known species of wild edible mushrooms, but they are known sources of food and income in more than 80 countries (*Boa, 2004*).

In Mexico, at least 371 macromycete species are traditionally consumed, making it the second country with the most species of wild mushrooms used as food, only after China (600 species), and it is the sixth country in the world with the highest number of ethnic groups (*Ruán-Soto, Garibay-Orijel & Cifuentes, 2006*; *Garibay-Orijel & Ruan-Soto, 2014*). The state of Oaxaca is one of the most biodiverse regions in the planet, and the most biologically and culturally diverse region in Mexico (*Flores-Villela & Gerez, 1994*), but there is a lack of mycological information for this area (*Garibay-Orijel et al., 2006*). The few studies on macromycetes in Oaxaca have focused on the functional diversity of macrofungal communities in the Costa region (*Caiafa et al., 2017*), taxonomy and traditional use of *Psilocybe* species in different localities of the state (*Guzmán et al., 2004*; *Ramírez-Cruz, Guzmán & Ramírez-Guillén, 2006*), the traditional use of macrofungi in the Mixteca region (*Santiago et al., 2016*), and the diversity and traditional use of macromycetes in the Sierra Norte region, which has the most complete inventory of useful macromycetes in Mexico, comprising a total of 159 taxa (*Garibay-Orijel et al., 2009*). Nevertheless, it is common knowledge that many communities throughout other regions of Oaxaca also use wild mushrooms.

However, it has been suggested that mushroom harvesting may affect macromycete communities and fruit body production in subsequent years by lowering spore-release, damaging mycelia, and disrupting biotic interactions with other species (*Arnolds, 1995*; *Leonard, 1997*; *Money, 2005*). Due to the role of macromycetes in ecosystem processes, and their nutritional and economic importance, concern about the negative effects of harvesting has grown among mycologists, conservation agencies, forest managers, landowners, and mushroom traders (*Boa, 2004*; *Leonard, 1997*; *Pilz et al., 2007*; *Pilz & Molina, 2002*). Nevertheless, results from experimental and long-term research have

indicated that over-harvesting causes no damage to the macromycete communities since only the fruit bodies are removed and the mycelium is left untouched (*Norvell, 1995*; *Egli et al., 2006*). Yet, soil compaction associated with mushroom collecting can reduce the number of fruit bodies per year (*Egli, Ayer & Chatelain, 1990*; *Egli et al., 2006*).

Therefore, researchers have recommended that collection of wild edible mushrooms should be regulated, and that rare/endangered species must be identified and protected from harvesting (*Leonard, 1997*; *Money, 2005*). In Mexico, the lack of official statistics and scientific knowledge on mushroom harvesting has caused the regulatory framework to be ambiguous, inconsistent, and difficult to comply with. The Wildlife Act, for example, considers the use of wild macromycetes, but if supported by a management plan with evidence showing that the extraction rate does not exceed the rate of natural regeneration, making it virtually impossible to obtain an official harvesting permit (*Benítez-Badillo et al., 2013*). Many rural communities of Oaxaca have community-based systems to decide upon and regulate forest management in their territories. Nevertheless, wild mushroom harvesting is frequently excluded from management plans due to the scarce information about the implications of this activity.

The present study was carried out in a community located in the highlands of the Mixteca region of Oaxaca, where people have been intensively harvesting wild edible macromycetes in the same places for many years. In spite of evidence from other regions of the world showing that over-harvesting causes no damage to macromycete communities, there is a widespread perception in Mexico that harvesting the same area for many years diminishes the number of species and fruit body production. These ideas, together with the lack of scientific information and the inconsistent regulatory framework regarding wild edible mushrooms, highlight the need for studies on how mushroom harvesting in different regions and ecosystems of this country may be affecting the structure of macromycete communities. The aim of this study was to assess differences on diversity and distribution of macromycete species between areas used for mushrooms harvesting and non-harvested areas, to infer about the potential effects of collection on macrofungal communities in the Mixteca region of Oaxaca. Since macromycete communities are susceptible to mycelium damage, reduction of spore release, and microclimatic/environmental variation, we predicted that the turnover of species composition between harvested and non-harvested sites would be conspicuous, and the likely changes of diversity and distribution of macromycetes along the study area would be more related to the variation of microclimatic/environmental factors than to the effect of harvesting.

## MATERIALS AND METHODS

### Study area and sites

The study was conducted in the community of Independencia (17°05′43″ N, 97°39′35″ W), which is part of the municipality of San Esteban Atatlahuca, in the Mixteca region of Oaxaca, Mexico. Independencia is found in the Sierra Madre del Sur mountain range at 2,670 m.a.s.l., in an area characterized by pine-oak forests. The climate is temperate subhumid with rains in the summer. Temperature ranges from 10 °C to 16 °C, and annual precipitation from 800 mm to 1,500 mm (*INEGI, 2008*).

With the assistance of local mushroom collectors, four study sites were defined in the communal forests of Independencia: two sites in areas where local residents harvest wild edible mushrooms, and two areas where no harvesting takes place. Sites 1 and 2 were established in non-harvested areas, and Sites 3 and 4 have been intensively harvested (all the fruit bodies in these areas were collected every 2 days during 7 months each year) for the past 9 and 5 years, respectively. We chose sites that were similar in their altitude (ranging from 2,560–2,700 m.a.s.l.), tree composition (dominated by one unidentified species of *Pinus* and two unidentified species of *Quercus*), topography of the terrain (hills with homogeneous surfaces lacking notable depressions or conspicuous areas of exposed rocks), and understory coverage (present and homogeneous along the study area). We tried to ensure environmental similarity between sites to avoid great differences that could mask the effects of harvesting on the variables we used to explain diversity and distribution variation. In each site we established 10 permanent plots of 10 m × 10 m located at least 10 m apart from each other, totaling a sampling area of 0.1 ha per site.

## Explanatory variables

Every sampling date we recorded the following microclimatic variables in each plot: air and soil temperature (°C), relative air humidity (%), water content in soil (%), and soil pH. Soil compaction was determined by calculating bulk density (gm/cm$^3$) and soil porosity (%). Since soil bulk density in the study sites ranged from 0.38 gm/cm$^3$ to 0.44 gm/cm$^3$, the soil texture for all sites was classified as sandy clay loam to clay loam. Other environmental variables recorded per plot included: slope (°), aspect (°), canopy openness (%), and moss, rockiness and bare soil cover (%). Leaf litter depth was measured at the beginning, middle, and the end of the sampling season. In each plot we counted and measured the diameter and height of all trees with a diameter >10 cm at 1.3 m above ground. Vegetation structure was characterized using the basal area (m$^2$ ha$^{-1}$), density (individuals ha$^{-1}$), and mean and maximum height (m) of the trees counted.

## Macromycete sampling

It was not possible to use molecular analyses to determine the number of species in our study area due to the lack of funding and limited access to suitable labs. For this reasons, we based species identification on the macro and micromorphological characters of fruit bodies. Prior to the sampling season, we obtained permission from the municipal authorities of San Esteban Atatlahuca to collect macromycetes in our study sites. Since mushrooms are ephemeral, samplings consisted in continuously collecting macromycete fruit bodies in the four sites every week. Macromycetes were collected only during the rainy season (June–October) of 2017, and the sampling procedure was the same for the four study sites in order to obtain comparable data useful to analyze how diversity varied along the studied area. Each site was sampled by the same person every week for 5 months, involving the same sampling effort (i.e., number of plots per site and sampling dates) in each place. To minimize the potential effect of collecting on future fruit body production, only one or two fruit bodies were collected per species for identification when necessary. Fruit bodies of the same species within a diameter <50 cm were recorded as a

single individual. To avoid soil compaction and raking leaf litter, samplings and data recording within the plots were carefully carried out by a single person. When specimens could not be identified at the species level, they were classified as morphospecies using a higher taxonomic level approach. Species were classified as edible or not based on information from local residents and a literature review (*Garibay-Orijel et al., 2009*; *Karun & Sridhar, 2017*).

### Data analysis

We recorded the number of macrofungal species in each site, and the observed species richness was compared between sites by means of rarefaction curves standardizing the samples to the minimum number of individuals recorded in one site. We constructed species accumulation curves to determine the effectiveness of the sampling effort (i.e., number of plots). We used analyses of variance (ANOVA) to determine differences between sites with regard to the number of individuals for each species, the number of species per site, and soil compaction. We used Tukey's HSD tests to identify pairs of means that differed from each other. Macrofungal diversity was calculated with the Shannon index, and with the true diversity index of first order (qD) using the multiplicative diversity decompositions of the effective numbers of species (*Jost, 2006*, *2007*). A single linkage hierarchical cluster analysis was performed based on composition and abundance of species. These analyses were conducted in R version 3.4.2 (*R Development Core Team, 2017*). The completeness of the macromycete inventories was estimated using the species richness estimator Jacknife 2, and the turnover of species composition was assessed with the Chao-Jaccard similarity index, both of which were calculated in EstimateS 9.1.0 (*Colwell, 2013*).

The Spearman correlation coefficient was calculated to determine the relationship between the explanatory variables and macrofungal richness. To understand the distribution of macromycete species with respect to our set of environmental, microclimatic, and vegetation structure variables, we used canonical correlation analyses (CCA). A lineal regression analysis was carried out to determine the relation between species similarity and geographic distance between sites. The *t*-test proposed by Hutcheson (*Zar, 2009*) was used to determine differences in Shannon diversity values between sites. Unless stated otherwise, statistical analyses were performed in R version 3.4.2. (*R Development Core Team, 2017*).

## RESULTS

### Macromycete species richness and taxonomic groups

We recorded a total of 856 individuals corresponding to 138 species, and 23 of these were identified as edible species. The phylum Basidiomycota was represented by 10 orders, 33 families, 59 genera, and 134 species; Ascomycota was represented by four orders, four families, four genera, and four species (Appendix A). Site 4 had the highest macromycete species richness (72), while Site one showed the lowest number of species (34). Similarly, the highest richness of edible species was found in Site 4 (14), and the lowest in Site 1 (9) (Table 1).

**Table 1 Macromycete diversity and abundance in the study sites.** Macromyctete species richness, estimated richness (Jacknife 2), diversity and abundance in each studied site of the Mixteca region of Oaxaca, Mexico.

| Site status | Site 1<br>Non-harvested | Site 2<br>Non-harvested | Site 3<br>Harvested | Site 4<br>Harvested |
|---|---|---|---|---|
| All macromycetes | | | | |
| Richness | 34 | 64 | 48 | 72 |
| Jacknife 2 | 65.14 | 118.07 | 84.5 | 139.67 |
| Shannon diversity | 1.17 | 1.54 | 1.33 | 1.53 |
| True diversity | 14.83 | 35 | 21.28 | 34.01 |
| Abundance | 115 | 221 | 177 | 306 |
| Edible macromycetes | | | | |
| Richness | 9 | 12 | 10 | 14 |
| Shannon diversity | 0.57 | 0.96 | 0.6 | 0.87 |
| True diversity | 3.7 | 9.08 | 4.02 | 7.47 |
| Abundance | 66 | 36 | 84 | 86 |

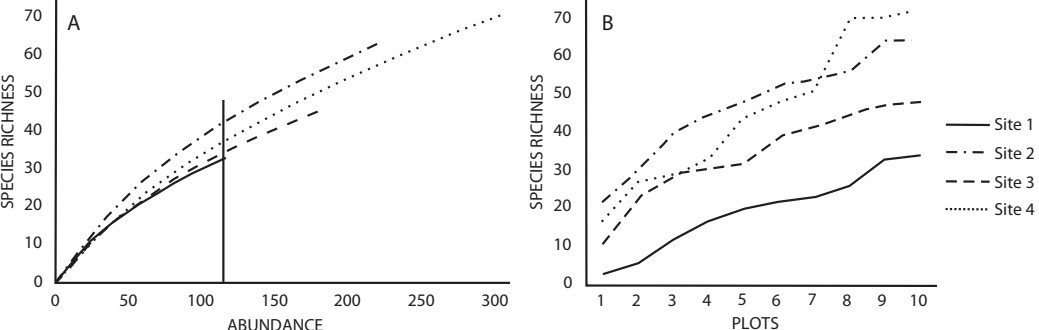

**Figure 1** **(A) Rarefaction and (B) accumulation curves for species richness in the four studied sites based on a standardized number of individuals and plots as sampling effort, respectively.** Vertical lines in rarefaction curves indicates species richness for the minimum number of individuals recorded in a study site.     

Similarly, the number of species estimated with the rarefaction curves (using a standardized abundance of 115 individuals) indicated that Sites 2 and 4 had the highest richness (42 and 38 species, respectively) compared to Sites one and three (33 and 35 species, respectively) (Fig. 1A). The rarefaction and species accumulation curves did not reach the asymptote, suggesting that our species inventories were not complete (Figs. 1A and 1B), however, the richness estimator Jacknife 2 indicated that the inventories were more than 50% complete. Variation of the estimated richness among sites corresponded to the variation of the recorded number of species (Table 1). The ANOVA for the abundance of each species indicated no differences between sites ($p = 0.87$), as well as the Tukey's HSD. The ANOVA for the number of species did indicate differences in species richness between sites ($p = 0.009$), but the Tuke's HSD test revealed that only Sites one and four differed significantly ($p = 0.017$).

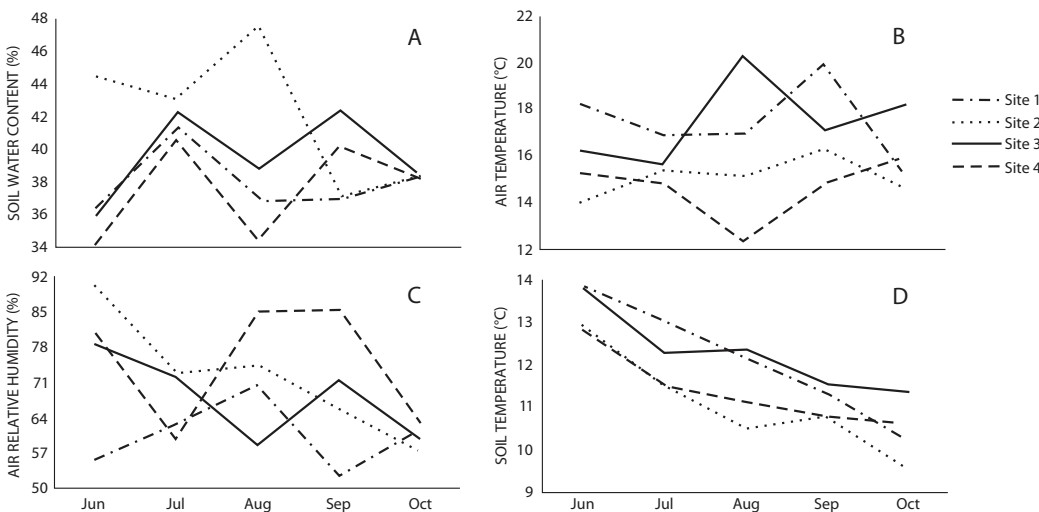

**Figure 2 Monthly variation of (A) soil water content, (B) air temperature, (C) air relative humidity, and (D) soil temperature in the studied sites through the sampling season.**

## Macromycete diversity and distribution

Both the Shannon and true diversity indices (Table 1) indicated that Site 2 had the highest diversity of macromycetes (1.54 and 35, respectively), and Site 1 had the lowest (1.17 and 14.83, respectively). The same patterns of Shannon and true diversity were observed for the edible species (Table 1), with Site 2 being the most diverse (0.96 and 9.08, respectively), and Site 1 being the least diverse (0.57 and 3.7, respectively). We found no statistical differences in Shannon diversity between Sites 1 (non-harvested area) and 3 (harvested area), and between Sites 2 (non-harvested area) and 4 (harvested area). The proportion of edible species with respect to the total of species recorded in each site was 26.5% for Site 1, 18.8% for Site 2, 20.8% for Site 3, and 19.4% for Site 4.

The microclimatic variables showed that air and soil temperature were higher in Sites 1 and 3, while relative air humidity was higher in Sites 2 and 4, and water content in soil was higher in Site 2 (Fig. 2). Spearman's correlation coefficient indicated that macromycete richness was positively correlated with relative air humidity, herbaceous plant coverage, slope, maximum tree height and tree basal area; and negatively correlated with air and soil temperature (Table 2).

The cluster analysis indicated that Sites 1 and 3 were similar in species composition, and Site 2 was similar to Site 4 (Fig. 3). Correspondingly, the Chao-Jaccard showed that for both the total macromycete species and the edible species, Sites 1 (non-harvested area) and 3 (harvested area) were the most similar, followed by Sites 2 (non-harvested area) and 4 (harvested area). Sites 1 and 4 were the most dissimilar in terms of total macromycete species, and Sites 1 and 2 were the most different with respect to edible species (Table 3). Geographic distance between sites and values of the Chao-Jaccard index were not significantly related ($p = 0.6$).

The CCA with microclimatic explanatory variables was carried out for 138 macromycete species considering air temperature, relative air humidity, soil temperature

**Table 2 Spearman correlation coefficients (ρ) between the species richness of macromycetes recorded in the studied area and the explanatory variables.**

| Variable | ρ | p-value |
|---|---|---|
| Air temperature*** | −0.58 | 0.00008 |
| Relative air humidity*** | 0.634 | 0.00001 |
| Soil temperature** | −0.414 | 0.007 |
| Water content in soil | 0.098 | 0.545 |
| Soil porosity | −0.004 | 0.977 |
| Soil pore space filled with water | 0.03 | 0.854 |
| Bulk density | 0.004 | 0.977 |
| pH | 0.14 | 0.388 |
| Litterfall | −0.068 | 0.676 |
| Rockiness | 0.169 | 0.294 |
| Moss cover | 0.214 | 0.184 |
| Herbaceous* | 0.336 | 0.033 |
| Slope* | 0.36 | 0.022 |
| Aspect | −0.099 | 0.541 |
| Canopy | 0.075 | 0.642 |
| Tree average height | 0.158 | 0.327 |
| Tree maximum height* | 0.372 | 0.017 |
| Tree basal area* | 0.329 | 0.038 |
| Tree density | 0.172 | 0.285 |

**Notes:**
* $p < 0.05$.
** $p < 0.01$.
*** $p < 0.001$.

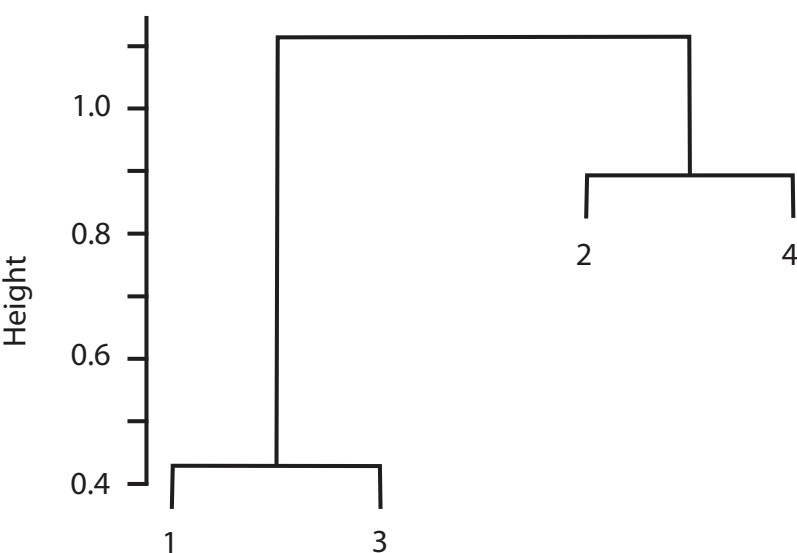

**Figure 3 Cluster analysis for the four studied sites, based on composition of species and abundance.** Euclidian distance is indicated by height values.

**Table 3 Chao-Jaccard similarity index between pairs of sites based on the composition of macromycete species.**

| Pairs of sites | All macromycetes | Edible macromycetes |
| --- | --- | --- |
| 1–2 | 0.7 | 0.17 |
| 1–3 | 0.79 | 0.88 |
| 1–4 | 0.55 | 0.53 |
| 2–3 | 0.69 | 0.33 |
| 2–4 | 0.73 | 0.74 |
| 3–4 | 0.64 | 0.65 |

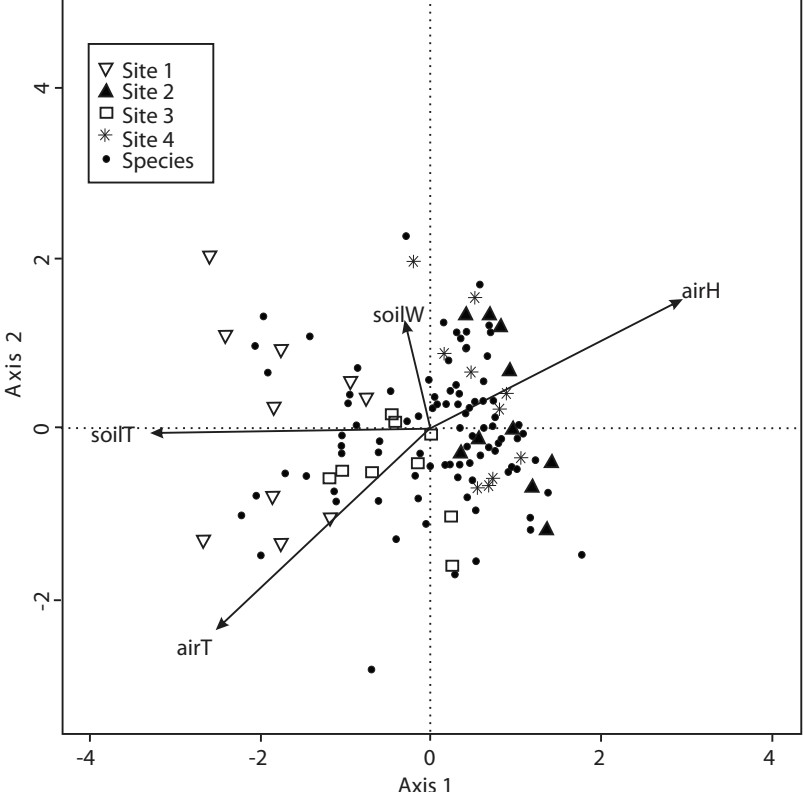

**Figure 4 CCA for all the recorded macromycetes in the four study sites.** Vectors are microclimatic explanatory variables: soil temperature (soilT), soil water content (soilW), relative air humidity (airH), and air temperature (airT).

and percentage of water in the soil. The model only retained air and soil temperature, but the other variables were included to better explain the ordination. Axis 1 (eigenvalue = 0.4926) and axis 2 (eigenvalue = 0.3226) accounted for 37% and 24% of the species-microclimate relationship, respectively. CCA results showed that Sites 1 and 3 were clearly separated from Sites 2 and 4 along the first canonical axis (Fig. 4).

The CCA for environmental variables was also carried out for the 138 macromycete species considering litterfall, canopy openness, slope, aspect, rockiness, moss and herbaceous coverage, bulk density, soil porosity and water-filled soil pore space. The model
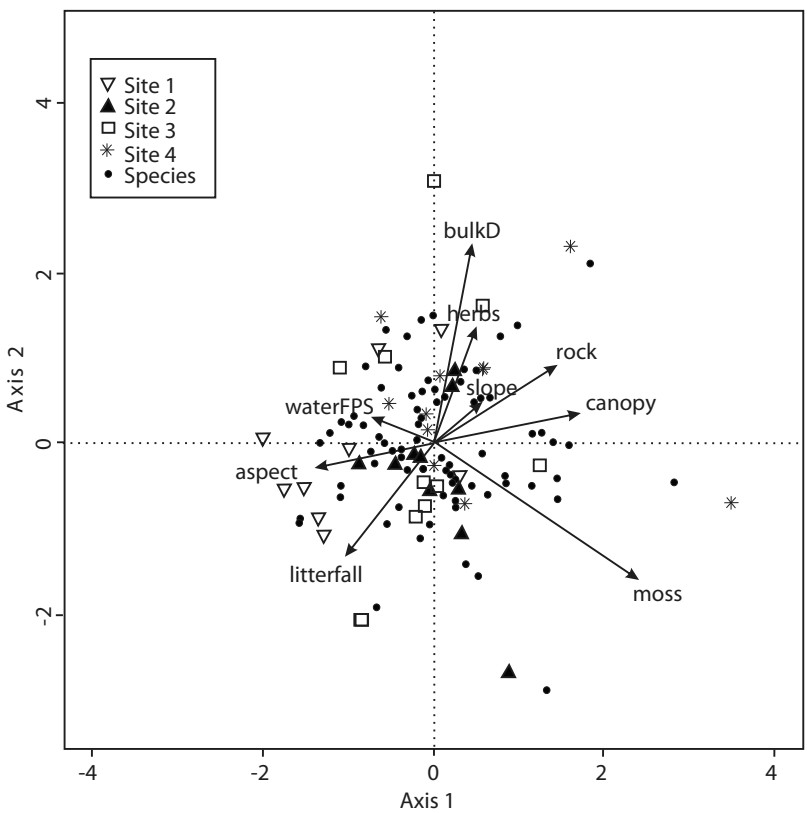

**Figure 5 CCA for all the recorded macromycetes in the four study sites.** Vectors are environmental explanatory variables: bulk density (bulkD), herbaceous coverage (herbs), rockiness coverage (rock), slope, canopy, moss coverage (moss), litterfall, aspect, and soil pore space filled with water (waterFPS).

only retained moss coverage, but the other variables contributed to explain the ordination. Axis 1 (eigenvalue = 0.4213) and axis 2 (eigenvalue = 0.3545) accounted for 17% and 14% of the explained species-environmental relationship, respectively (Fig. 5).

The CCA for vegetation structure also considered 138 species and used mean tree height, maximum tree height, tree basal area and tree density. The model only retained maximum tree height, but the other variables were included to explain the ordination. Axis 1 (eigenvalue = 0.4509) and axis 2 (eigenvalue = 0.3049) accounted for 38% and 25% of the explained species-vegetation structure relationship, respectively. CCA results showed that Sites 1 and 3 were separated from Sites 2 and 4 along the first canonical axis (Fig. 6).

## DISCUSSION

Mexico is one of the main consumers of wild edible mushrooms in the world. Different studies have shown the high diversity of these organisms in the country and their importance as sources of food and income for human communities in rural areas. For instance, in a forest of La Malinche National Park in Tlaxcala, 93 macrofungal species were recorded, 91 of them reported in the literature as edible, and 74 species were found to be used by the local people (*Montoya et al., 2004*). In Ixtlan, Oaxaca, 159

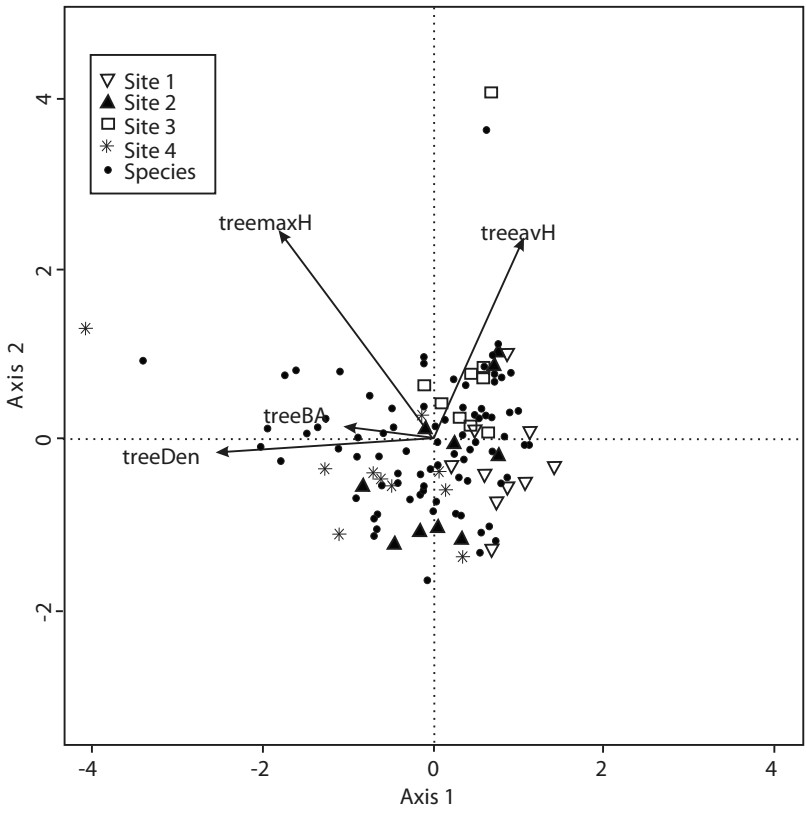

**Figure 6 CCA for all the recorded macromycetes in the four study sites.** Vectors are vegetation structure explanatory variables: tree maximum height (treemaxH), tree average height (treeavH), tree basal area (treeBA), and tree density (treeDen).

macromycete taxa were reported as having a use, including 113 edible species (*Garibay-Orijel et al., 2009*). In the Sierra del Ajusco, in Mexico City, 29 wild edible species were found in just 800 m² (*Zamora-Martínez & Nieto de Pascual-Pola, 1995*). In Cerro El Zamora, located between Guanajuato and Queretaro, a study identified 130 macromycete species, 55 of which were recognized as edible based on a literature review (*Landeros et al., 2006*). In our study area, a total of 138 macromycete taxa were recorded, and 23 were recognized as edible according to a literature review. Interviews with local people, performed as part of an ethnomycological study conducted simultaneously to this one, showed that they consume at least 45 species. Despite the fact that most studies on macromycetes comprise 1–3 years of sampling, it has been suggested that 5 to 10 years would be more suitable (*Lodge et al., 2004*), but it is rarely possible to sample over the many years required to document most of the species (*Gabel & Gabel, 2007*). Our sampling period spanned only 1 year, however, the aim of this study was not to obtain complete macromycete inventories but to assess the diversity variation among harvested and non-harvested areas, so we used the same systematized procedure in the four study sites (see Materials and Methods) to get comparable data. Although our permanent plots in the harvesting sites were marked with barricade tape and it was agreed with the local population that they would not collect mushrooms within the plots during the

sampling season, traces of harvesting were often found. Locals ensure that they only collected *Tricholoma mesoamericanum* in our plots, which could explain why the species was not recorded by us.

The diversity of macromycetes did not differ between the non-harvested (Sites 1 and 2) and harvested (Sites 3 and 4) areas. Both for all macrofungi and for the edible species, Sites 2 and 4 showed a similar diversity and were the most diverse, while Sites 1 and 3 were similar and less diverse. It is broadly known that macromycete communities are strongly influenced by habitat heterogeneity and microclimatic variation, and our results on diversity can be clearly explained by the observed microclimatic conditions. In spite of having similar environmental conditions and vegetation structure in the four studied sites, microclimate was not homogeneous throughout and the differences corresponded to the observed patterns of diversity where Sites 2 and 4 on the one hand, and Sites 1 and 3 on the other, were more similar to each other in terms of the microclimate and macromycete diversity. Several studies have suggested that humidity, precipitation and temperature are the main factors affecting macromycete fruiting and diversity in both temperate and tropical forests (e.g., *O'Dell, Ammirati & Schreiner, 2000*; *Ohenoja, 1995*; *Lodge et al., 2004*; *Brown, Bhagwat & Watkinson, 2006*; *Durall et al., 2006*; *Gómez-Hernández et al., 2012*), and that temperature and humidity are the best predictors for fungal richness (*Talley, Coley & Kursar, 2002*). Our results showed that air and soil temperatures were higher in Sites 1 and 3, and negatively correlated with macromycete species richness. Likewise, relative air humidity was higher in Sites 2 and 4, and positively correlated with species richness. These results suggest that mushroom harvesting is not likely affecting the assemblages of edible macromycetes, nor disturbing environmental factors of relevance for macrofungal communities. This is consistent with different long term studies evaluating the effect of mushroom harvesting on the number of macromycete species and fruit body production. In a 29 year study carried out within two fungus reserves in southwestern Switzerland, systematic harvesting was applied using picking and cutting techniques and the results indicated that regardless of the harvesting technique, neither macromycete species richness nor fruiting were affected (*Egli et al., 2006*). Similarly, 13- and 40-year studies conducted in the United States and Sweden, respectively, revealed that intensive collecting of wild mushrooms did not reduce annual production of fruit bodies (*Jahn & Jahn, 1986*; *Norvell, 1995*). It has been suggested that stability in the number of macromycete species and fruiting in areas under harvesting pressure may be explained by the hundreds of spores released from each fruit body before and during mushroom collection, or because enough spores disperse from adjacent areas (*Money, 2005*; *Egli et al., 2006*).

Apart from microclimatic conditions, numerous environmental variables have been related to macrofungal diversity and fruit body production, such as slope, aspect, basal area, presence of rocks, and density of trees (*O'Dell, Ammirati & Schreiner, 2000*; *Ferris, Peace & Newton, 2000*; *Cavender-Bares et al., 2009*; *Egli et al., 2010*; *Gómez-Hernández et al., 2012*). Our results showed a positive correlation between slope and the number of macromycete species, agreeing with findings by *Caiafa et al. (2017)* in the Costa region of Oaxaca. But understanding how the slope influences macromycete richness can be a

difficult task due to the variety of biotic and abiotic factors related to the soil environment. Findings have suggested that the slope effect on macromycetes is related to vegetation type, as well as to the moisture and temperature gradient along the slope. However, there are discrepancies between studies since some of them report a positive relation between slope and species richness, and others report it to be negative (*Nantel & Neumann, 1992*; *Rubino & McCarthy, 2003*; *Gómez-Hernández et al., 2012*). In this study, basal area and maximum height of trees were positively correlated with species richness, and a greater amount of fruit bodies were recorded in areas with wider and taller trees. Correspondingly, the highest basal area, maximum height of trees, and macromycete richness and abundance were recorded in Site 4. Related studies have proposed that the composition and structure of host tree communities can influence macromycete richness and fruit body production by affecting fungal specialization and providing different habitats and resource quality and quantities (*Villeneuve, Grandtner & Fortin, 1989*; *Richard et al., 2004*; *Brown, Bhagwat & Watkinson, 2006*; *Zhang et al., 2010*). In our study, herbaceous cover was positively correlated with macromycete species richness, agreeing with results that suggest a trend towards increasing the number of macromycete fruit bodies with increased presence of herbaceous plants, and a positive relation between the number of macromycete species and fruit body production (*Mehus, 1986*; *Toledo, Barroetaveña & Rajchenberg, 2014*). The observed trend can be explained by the fact that the herbaceous layer provides up to 16% of annual litter fall and influences the cycling rates of N, P, K and Mg, which are important nutrients for fungal growth and health (*Gilliam, 2007*). Soil compaction by trampling has been proposed as one of the consequences from harvesting that can trigger a decrease in macromycete diversity and fruit body production by causing mycelium smashing (*Arnolds, 1995*; *Watling, 2003*). *Egli, Ayer & Chatelain (1990)* intensively trampled a plot every 2 days during summer and autumn for 1 year, and observed a strong decrease in fruit body production. People in our study area harvest mushrooms every 2 days for 7 months every year, however, the soil water content (which is directly related to soil compaction) was similar between non-harvested and harvested sites, and macromycete abundance was higher in the harvesting sites. Also, the ANOVA for soil compaction showed no differences between sites. These results suggest that trampling due to mushroom collection has not caused severe soil compaction and damage to the macofungal communities despite many years of intensive harvesting.

The sites assessed are covered by pine-oak forests with marked dominance of pines, and the composition of tree species was similar in all sites. In forests with low diversity of tree species, as in our study, the opportunity for macromycete specialization increases due to the high abundance of few tree species, and the composition of specialist fungi has been observed to change across the distribution of a vegetation type (*Nantel & Neumann, 1992*; *Ferrer & Gilbert, 2003*; *Lodge et al., 2008*). The turnover of macromycete species between our four study sites was not as conspicuous as expected. The similarity in species composition ranged from 55% to 79%, and resembled the trend observed for microclimate since it was similar between Sites 2 (non-harvested) and 4 (harvested), and between Sites 1 (non-harvested) and 3 (harvested). Corresponding with our results, other

studies have reported that variation of macrofungal species composition between sites, within a same vegetation type, was more related to precipitation and temperature than to the composition of tree assemblages (*Marmolejo & Méndez Cortes, 2007*; *Cavender-Bares et al., 2009*; *Gómez-Hernández & Williams-Linera, 2011*). Furthermore, the ordination analyses indicated that air and soil temperature, relative air humidity, and the humidity-related variables of moss coverage and maximum tree height were the main factors involved in the distribution of macromycetes throughout our studied area. Other environmental and vegetation structure variables were homogeneous in the four studied sites and equally related to macromycete distribution, thus they did not play a key role in the observed changes in species composition between study sites. In accordance with our results on diversity and species richness, our findings suggest that microclimatic differences best explained the differences in macromycete distribution along the studied area.

In order to avoid pseudoreplication issues, we had ten plots (subsamples) in each study site (replicates) for data collection, and these were analyzed as independent samples. Nevertheless, it was only possible to establish four sites in the study area, thus two replicates were assigned to each treatment (i.e., harvested and non-harvested). In spite of having more than one observational unit for each treatment, the low number of replicates could result in an underestimation of the variability in the treatments. Our results correspond to others reported in several previously mentioned studies, but it would have been valuable to include a larger number of study sites in each treatment to make our results more robust.

## CONCLUSIONS

This study has shown that harvesting wild edible mushrooms for several years within a specific area may not represent a threat to macrofungal communities, and it can be a sustainable activity. Patterns of diversity and distribution of macromycetes along harvested/non-harvested areas are mainly determined by the intrinsic microclimatic variation between sites. The present study included only one season of data, which could be a limitation to capture long-term differences. Thus carrying out long term studies on different ecosystems and evaluating harvesting techniques is of great interest to elucidate the most suitable methods to best manage this valuable non-timber forest product. Surveys along disturbance gradients are also desirable to clearly determine whether harvesting wild mushrooms is an innocuous activity as long as the general environment and macromycete habitat are not disturbed. Generating more information on this activity will allow improving regulatory frameworks and not to exclude mushroom harvesting from management and conservation plans.

## ACKNOWLEDGMENTS

We thank Dr. Virginia Ramírez Cruz and M.Sc. Alonzo Cortés Pérez for their help with macromycete identification in the field and herbarium; Dra. Sandra Smith Aguilar for helpful suggestions and comments on earlier versions of this manuscript; Raúl Rivera García for his help with the figures edition.

### Funding

Carolina Ruiz-Almenara was supported by the CONACyT scholarship No. 611195. There was no additional external funding received for this study. The funders had no role in study design, data collection and analysis, decision to publish, or preparation of the manuscript.

### Grant Disclosures

The following grant information was disclosed by the authors:
CONACyT scholarship: 611195.

### Competing Interests

The authors declare that they have no competing interests.

### Author Contributions

- Carolina Ruiz-Almenara performed the experiments, analyzed the data, prepared figures and/or tables, approved the final draft, carried out field work.
- Etelvina Gándara conceived and designed the experiments, performed the experiments, contributed reagents/materials/analysis tools, authored or reviewed drafts of the paper, approved the final draft.
- Marko Gómez-Hernández conceived and designed the experiments, performed the experiments, analyzed the data, contributed reagents/materials/analysis tools, authored or reviewed drafts of the paper, approved the final draft.

### Field Study Permissions

The following information was supplied relating to field study approvals (i.e., approving body and any reference numbers):
Municipal authorities of San Esteban Atatlahuca gave collecting permission.

### Data Availability

The raw measurements are available in the Supplemental Files.

### Supplemental Information

Supplemental information for this article can be found online at http://dx.doi.org/10.7717/peerj.8325#supplemental-information.

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
