# Peer review of "Comparison of diversity and composition of macrofungal species between intensive mushroom harvesting and non-harvesting areas in Oaxaca, Mexico"

_PeerJ, doi:10.7717/peerj.8325_

## Round 0.1 · original submission · Major Revisions

The editor apologizes for the long time, it required to get the manuscript reviewed. It was, as usually in summer, very difficult to obtain reviewers. In this case I had to ask over 20 colleagues to obtain 2 reviews. The reviews of the manuscript were pretty divergent in their conclusions. I looked throught the comments and would think that the comments of reveiwer 2 are addressable and recommend, therefore, a major revision. The paper might be sent back to the reviewers or to a new set of reviewers.

I would like to, especially, point out a few points from the reviewers that I deem important. Firstly, bot reviewers point out that the description of the sites (stand characteristics, soils etc....) is not up to standards. Secondly, the reviewer 1 claimed that the paper is likely of low scientific importance. Scientific importance is not a selection criteria for PeerJ but it would be good if the authors strengthen what is new and important in the paper.

The English of the paper is fair but not up to publication standards. I would recommend the authors to have it revised if they do not want to risk significant delays in publication.

A few other comments:
1) Intensive harvesting: It would be good to know what means intensive harvesting since this will be very different for different locations in the world. If you can give evaluations on how often sites are harvested it would be good.

2) The paper has an issue with pseudoreplication because there are only 4 sites. I think that pseudoreplication is often inevitable in many ecological studies because unharvested sites etc are rare. I, however, do think that authors should be honest about pseudoreplication and possible implication for inference. Please include a paragraph on this in the discussion.

Reviewer 1 ·

Basic reporting

There are no page numbers in the manuscript.

There were a few typos and some information was lacking from the literature references. I would recommed to check there carefully.

Experimental design

One season study for mushroom fruitbody communities is not able to capture all or not always the majority of the species (see the non-saturating curves of rarefaction and accumulation is figure 1.). This may have an impact on the results and therefore it should be clearly stated both in methods and discussion why the results are presented based on 1 year sampling because most studies on this topic are based on 2-3 (or more) years inventories. It is also important to note the reader that the rarefaction curves do not saturate.

There was unintentional harvesting by the local people. This is an important point as it may have an impact on the results and it should be more clearly stated in the manuscript.

Aboveground mushroom community is not the same as the belowground community. More information of the community could be achieved by doing molecular analyses from the soil. I would like to see justification in the introduction/matmet why the authors decided to carry out this project based on mushrooms only.

Validity of the findings

The results indicated that harvesting did not have an impact on the fruitbody communities but that environmental variables were able to explain the findings to some extent. Although I consider the scientific value of this manuscript not be be very high, I still would like to see it published after revision. There are relatively few studies on mushroom fruitbody communities. Therefore I think it would be good to get publications on this topic.

Additional comments

The authors hypothesized that harvesting does have an impact on the mushroom yields. However the results indicate that harvesting did not have an impact but that the weather and environmental conditions play more important role. This is what can be expected: mushroom fruitbody yields are very dependent on environmental conditions. Especially the different weather conditions may cause notable differences between years. In addition, I am especially concerned if the results were affected by unintentional harvesting of “the non-harvesting treatment plots” which was suggested in the discussion. I think this is an important point and should be mentioned already in the Materials and methods.

Reviewer 2 ·

Basic reporting

Sentences could be improved in English. References cited in the Introduction are relatively old. Raw data are available but not very well annotated.

Experimental design

The content of the paper is within the scope of the journal, but does not very well correspond to the title of the paper. The design for the field investigation is ok, but the data obtained are not dependable. Pl. see the problem with methods in the comments.

Validity of the findings

Pl. see above.

Additional comments

1. Lines 44-51: Since the article mainly focuses on mushroom harvesting and its effects on macromycetes, it is suggested to add some information on the relationship between mushroom and macromycetes but not mycorrhizal fungi and pathogenic fungi.
2. L62-63: it is suggested to delete the sentence since it is about eating culture and habit, which may influence on mushroom harvesting, but not directly be related with the topic of the article.
3. L86-99: it is suggested to simplify this paragraph. For example, using one sentence to outline the paragraph and having it integrated into the previous paragraph.
4. It is suggested to restructure the paragraph of L100-110. Purposes of composing this paper should be stated clearly, but not the predictions.
5. The references cited in the Introduction are restively old. It is suggested to use as many latest references as possible.
6. L114-121:since no specific village name is mentioned in the paper, it is suggested to change village to four sites. “surrounding Independencia” means not in or inside the Independencia.
7. L133-134: Since mushrooms are ephemeral, sampling conducted every week seems to have a too long interval. Was sampling not done on a daily basis? Or only the calculation of sample quantity was conducted on a weekly basis?
8. L123-127: it is stated that the characteristics of the selected sites were as similar as possible in terms of altitude, tree composition, vegetation structure, topography of the terrain, and understory coverage. It is suggested to have a more detailed description of these aspects.
9. L147-149: please specify the bulk density of the four sites, not the criteria for the division of the bulk density.
10. Since a paper is composed of written sentences, but not oral sentences, it is suggested to use as many written sentences as possible. For example L151-153.
11. For explanatory variables, no specific description is available in the paper and readers have to read raw data / supplementary files. It is suggested to have a summary description for each variable. No title is available for supplementary files 2 and 3 and no unit for each variable in the file 3 although they are supplementary files.
12. It is suggested to describe explanatory variables first, followed by macromycete sampling.
13. Appendices A and B could be combined into one, with edible species marked with asterisks in Appendix A. Latin names of some species in the appendices have no space between two words. Pl. add notes to the appendices to explain sp. 1, sp. 2, etc. Do they mean species unidentified?
14. L179-180: are Basidiomycota and Ascomycota two extremes of species richness? Description is not very clear.
15. In L155-166, it is stated that both effectiveness of the sampling effort and the completeness of the macromycete inventories were estimated. So other data analyses should be based on the estimation of effectiveness of sampling and completeness of inventories. As stated by authors, the effectiveness of sampling efforts and completeness of inventories were not as good as expected, validity of the data is questionable.
16. Since the topic of this paper is to study the effect of intensive mushroom harvesting, harvesting intensity (for example, weekly harvesting quantity) should be mentioned in the paper, but nothing is read. According to the paper, only are harvesting and non-harvesting described. In addition, according to the design, analysis of variance could be performed to see whether there is significant difference between harvesting and non-harvesting, based on which further detailed analysis of reasons behind the phenomenon could be continued. A richness-based result and clustering of species composition indicating no influence of harvesting are not dependable.

Annotated reviews are not available for download in order to protect the identity of reviewers who chose to remain anonymous.

---

## Round 0.2 · Minor Revisions

I looked again through the manuscript, and I was positively surprised by the changes. I think the study looks now and makes a contribution to our understanding of tropical mountain fungi

I have three points to add:

1) Sampling and statistical analysis: You sampled the 4x10 plots 5 x during the study. It looks like some analyses (like the CCA were d
on this dataset consisting of 40 observations. However, I suspect that the Spearman correlation uses a larger dataset of 4x10x5 observations (i.e. each sampling time is treated as an independent observation). This is ok if the authors clearly mention that this temporal autocorrelation.

2) The authors outline some possible causes for changed species abundance (like soil compaction), litter raking and vegetation disturbances. Would it be possible to quantify these effects by, for example, an ANOVA or harvested and non harvested sites? (I think, even if harvesting does not cause soil compaction it is important to prove it)

3) A naive question since I do not know the sites and the species. How, did the sampling work. When harvesting mushrooms (I do it for culinary purposes not for science) I usually come along small fruiting bodies and it is difficult to find all fruiting bodies. I could think that this is even more difficult in a Mexican pine forest with dense understory than in a Finnish open forest. How did you assure that you have all fruiting bodies or is there any protocol (like sampling time)? Did you have a minimum size for fruiting bodies?

---

## Round 0.3 · accepted · Accept

I am looking forward to see your manuscript published. I apologize again for the long delay in the first revision but I had to ask an exceptionally large number of colleagues to get two reviewers committed.